# Evaluation of Frost Damage and Pod Set in Faba Bean (*Vicia faba* L.) under Field Conditions

**DOI:** 10.3390/plants10091925

**Published:** 2021-09-15

**Authors:** Najeeb H. Alharbi, Salem S. Alghamdi, Hussein M. Migdadi, Ehab H. El-Harty, Kedar N. Adhikari

**Affiliations:** 1IA Watson Grains Research Centre, School of Life and Environmental Science, The University of Sydney, Narrabri, NSW 2390, Australia; nalsubhi@kacst.edu.sa; 2The Life Science and Environment Research Institute, King Abdulaziz City for Science and Technology, Riyadh 11442, Saudi Arabia; 3The National Center for Vegetation Cover Development and Combating Desertification, Riyadh 11195, Saudi Arabia; 4Legume Research Group, Plant Production Department, College of Food and Agricultural Sciences, King Saud University, Riyadh 11362, Saudi Arabia; salem@ksu.edu.sa (S.S.A.); Hmigdadi@ksu.edu.sa (H.M.M.); eelharty@ksu.edu.sa (E.H.E.-H.); 5National Agricultural Research Center, Baqa, Amman 19381, Jordan

**Keywords:** pod set, radiation frost, advection frost, frost tolerance, sowing dates

## Abstract

Frost is one factor that causes extensive yield losses globally. A study was conducted to evaluate frost damage under field conditions and assess the genetic variation of flowers converting into pods. Diverse faba bean genotypes were evaluated under four growing seasons in a randomized complete block design: three at the University of Sydney, Narrabri, Australia (2014–2016) with three sowing dates, and one at the Agricultural Research Station, Dirab, Riyadh, Saudi Arabia (2016/2017) in one sowing. Visual methods were used to estimate frost damage and record the development of pods. Radiation frost in 2014 (Narrabri) damaged lower pods, while advection frost in 2016/2017 (Dirab) damaged upper pods. The radiation frost formed immediately above the ground; therefore, flowers and pods of taller plants minimized the damage because of their long distance from the ground. The earliest (mid-April) and middle sowing (7 May) suffered more by frost, while a delay in sowing (last week in May) led to frost escape or minor damage. The genotypes IX474/4-3 and 11NF010a-2 showed low sensitivity to frost at the vegetative and reproductive stages. Flowers developed at the beginning of flowering had a faster and higher pod formation rate (41–43%) than those formed later and contributed more to yields. Therefore, a severe frost at the beginning of flowering can cause a significant yield loss as these flowers are the most productive. The frost-tolerant genotypes, and faster and higher pod forming rates, identified in this study can be exploited to breed better varieties in the future.

## 1. Introduction

Faba bean (*Vicia faba* L.) is a cool-season grain legume grown in most parts of the globe for food and feed purposes. It has high yield potential, but environmental factors resulting in unstable yield highly affect its growth. Frost is one of the significant abiotic stresses threatening faba bean production [1,2,3]. Economic loss due to frost in Australia is estimated at $120–700 million each year [4,5]. Crops lose at least 10% yield because of the direct effect of frost in Queensland and northern New South Wales [6], but in severe frost, as much as 85–100% yield loss was reported for wheat (*Triticum aestivum* L.) in some areas [6,7].

The level of frost damage is determined by several factors, such as crop sensitivity, growth stage, frost severity, duration, and frequency [8]. Frost can be divided into two categories: radiation frost (calm night, clear dry atmosphere and no wind) and advection frost (windy day or night with temperature below freezing point) [9]. Radiation frost is the most common, including in Australia [10], while both types are common in Middle Eastern countries such as Jordan and Saudi Arabia [11]. According to Dalezios [8], radiation frost is less harmful than advection frost. Protecting crops from radiation frost in a small area may be possible by mixing the thin layer of cool air immediately above the ground with warmer layers to disperse the frosty conditions [12] using wind machines such as fans [9,13]. However, this is not possible in extensive areas.

Frost tolerance at the vegetative stage has no association with tolerance at the reproductive stage [6,14]. Flowering is the most sensitive time for frost in crops, including faba bean [15,16]. However, frost at flowering in the determinate crops (e.g., wheat) is more harmful than in the indeterminate crops (e.g., faba bean). Pods of faba bean are more tolerant than flowers, because their thick wall protects the developing ovules [17].

Improving frost tolerance in cool-season food legumes is fundamental for winter growing [18,19]. Understanding their responses to temperature stresses will help to improve the tolerance of legumes and help to secure the global food supply [20].

Faba bean produces a high number of flowers, 50–80 per plant, which equals 13.5 million flowers per hectare [21,22], but only 8–20% are converted into pods [22]. Other legumes, such as field bean *Lablab purpureus* (L.) sweet and chickpea (*Cicer arietinum* L.), also produce excessive flowers, but the success in converting flowers to pods is much higher in chickpea (*Cicer arietinum* L.) at 50–80% [23]. Failure in the reproductive organs, e.g., lack of pollination or fertilization resulting in flower or pod abortion, is a significant factor in the unsteady yield in grain legumes, including faba beans [24].

In addition, pods differ in their contribution to final yield based on their time of formation and position on the stem [25]. A faba bean study found that the weight of 1000 seeds in the lower position (665 g) was higher than in the middle and upper positions (627 and 541 g, respectively) [26]. In addition, Togun and Tayo [27] demonstrated that 99% of the matured pods in pigeon pea (*Cajanus cajan* (L.) Millsp.) were from flowers that opened in the first 24 days of the flowering. The early forming pods in pigeon pea were also usually heavier than those forming later [27], because the longer filling duration allowed more time to develop pods and fill seeds [28].

This study aimed to (1) identify the variation of frost tolerance in selected faba bean genotypes at different growth stages, (2) investigate the role of sowing times in managing frost damage, and (3) illustrate the genetic variation of proportion of flowers converting to pods.

## 2. Materials and Methods

### 2.1. Experimental Site and Plant Materials

This study evaluated frost damage under open field conditions over four growing seasons: three at The University of Sydney, Narrabri, Australia (30°16′26.7″ S 149°48′34.0″ E), with three sowing dates (16 April, 7 and 26 May in 2014, 17 April, 7 and 28 May in 2015 and, 29 April, 12 and 31 May in 2016). At Dirab, the Agricultural Research Station in the South of Riyadh, Saudi Arabia (24°25′17.7″ N 46°39′11.8″ E), one sowing date, 26 October 2016, was applied. 

A randomized complete block design (RCBD) was used in all growing seasons. The plot size varied in each season: 1 m × 6 m in 2014 (two rows, 50 cm apart), 2 m × 10 m in 2015/2016 (four rows, 50 cm apart), and 2 m × 3 m in 2016/2017 (four rows, 50 cm apart). The plant density for all experiments was 20 plants per m^2^. Seeds were sown using a tractor-mounted tine seeder at Narrabri and by hand at Dirab (5 to 10 cm depth). The soil type at the experimental site was vertosol clay at Narrabri and sandy clay loam at Dirab.

Ninety-four genotypes were evaluated in this study in 2014. In 2015 and 2016, only 15 genotypes were used, six from the 2014 experiment and nine with varying frost tolerance from the faba bean breeding program at Narrabri. The identical 15 genotypes, along with an additional local variety (Giza Blanka, a major type of “broad bean”), were used at Dirab in 2016/2017. 

### 2.2. Environmental Conditions

Minimum and maximum temperatures and the total amount of rainfall and irrigations were recorded. Data for the 2014 season were taken from Narrabri Airport Automatic Weather Station (AWS) (www.bom.gov.au/climate, accessed on 13 June 2015), while in 2015, 2016, and 2016/2017, the weather data were recorded using the data logger at the site (Figure 1). Frost events were recorded during the growing seasons. Frost occurred 14 times in 2014, seven times in 2015, and four times in 2016 (Figure 2). During the experiment at Dirab from October 2016 to February 2017, advection frost occurred twice on 4th and 5th February, reaching −4.19 and −0.6 °C, respectively.

### 2.3. Data Collection

#### 2.3.1. The Field Experiments from 2014 to 2016 at Narrabri

Frost damage was estimated by different methods, depending on the circumstances during each growing season. A visual 1–9 scale (1: no symptoms; 9: highly damaged plants/flowers/pods) was used for determining frost damage on the vegetative and reproductive organs by examining the stems (bending) and pods (blackening) in 2014 and the bending stems and damaged flowers in 2015 and 2016. At the seedling stage in 2015, the frost damage was estimated by counting the number of damaged plants per 20 m^2^. The effect of frost on pod sets was evaluated in 2015 by tagging: five random plants per plot and nodes with at least three flowers per plant affected by frost. Three points were considered while selecting flowers for tagging, which are: standard flowers (fully open), minimum of three flowers per plant (five plants per plot), and if the selected flowers were from a frost-damaged stem, the selection was made below the bending point to avoid any effects of bending.

In 2016, damaged plants per 20 m^2^ at the seedling stage could not be counted, because frost did not occur at this stage. Tagging of the damaged flowers after a frost event was also not possible because heavy rain followed the frost for the next three days.

#### 2.3.2. The Field Experiment in 2016/2017 at Dirab (Saudi Arabia)

Frost damage on stems and pods was scored using the visual 1–9 scale as described above. Regarding evaluation of the pod set, four plants with at least two flowers each (eight flowers per 6 m^2^) were randomly selected and tagged using paper tags to measure their pod conversion percentage and time taken to form pods. The first tagging occurred at 50% flowering (50 days after sowing (DAS) on December 14 (Tag_1_)) and then every two weeks. The total number of tagged flowers each time was 384 flowers (eight flowers × 16 genotypes × three replications). Mild temperature was observed between tags 1 and 3, from 14 December 2016 to 23 January 2017, but the temperature dropped, and a harsh frost (−4.19 °C) occurred 11 days after the fourth tagging (4 February 2017), resulting in the abortion of all tagged flowers in this group; therefore, the percentage and duration of flowers’ conversion into pods was recorded from Tag_1_, Tag_2_, and Tag_3_ only (Figure 3). 

### 2.4. Statistical Analysis

Data were analyzed by analysis of variance (ANOVA) using the Residual Maximum Likelihood (REML) function in GenStat 17th Edition (VSN International). A two-way ANOVA was conducted (genotypes, sowing dates, and their interaction), with replication to control for frost damage, in 2014, 2015, and 2016. In 2016/2017 (Dirab), a one-way ANOVA was used (genotypes) because the experiment covered a single season. Two-way ANOVA was conducted for pod set study where the proportion of flowers converting into pods was determined (genotypes, time of tagging flowers (flowering stage), and their interaction). No yield data were collected from Dirab. The difference in means (in all growing seasons) was identified by using the least significant difference (l.s.d.) at the ±5% level of probability. Graphs were prepared using GraphPad Prism 7. Correlation coefficients were obtained using Pearson’s test. 

## 3. Results

### 3.1. Response of Genotypes to Frost in 2014 at Narrabri

Fourteen frost events occurred in this season: nine in July, four in August, and one in September. The harshest frost was recorded on 12 August, when plants were exposed to freezing temperatures for eight hours; the minimum temperature was −3.9 °C. Damage was observed on stems and pods. The frost damage on the lower pods was significant. 

The 2014 experiment included several unadapted genotypes and preliminary breeding lines. Therefore, the variation was high among genotypes for stems (0.8 *** (*p*-value < 0.001)) and pods (0.7 ***). Variation was also found among sowing dates in stems and pods (0.14 * (*p*-value < 0.05) and 0.13 ***, respectively). Frost damage was decreased by the delay in sowing time. The interaction of genotypes with sowing dates (G × S) was not significant for frost damage on stems and pods. The sensitivity of stems and pods to frost showed high variability among the genotypes on all sowing dates. Comparison to Fiord (check for high frost sensitivity), four sister genotypes (11NF003a-11, 11NF003a-16, 11NF003b-1 and 11NF003e-4) that were selected from one cross (354g/1-4 × 114/1-16), showed high frost tolerance along with IX585c/1-2 on stems. Regarding frost damage on pods, only IX585c/1-2 showed low frost sensitivity in pods. 11NF003a-16 showed low sensitivity on stems but high sensitivity on pods, while IX585c/1-2 showed low sensitivity on both stem and pods. 

### 3.2. Frost Damage in 2015 at Narrabri

Seven frost events occurred during the 2015 growing season: one each in May, June, and August, and four in July. Damage was monitored after each occurrence. The severity of frost was less than in 2014; damage occurred only on leaves, stems, and flowers, but not on pods.

#### 3.2.1. The Number of Damaged Plants at the Seedling Stage

The symptoms of frost damage appeared seven days after the frost occurrence. Certain stages were less sensitive than others. For example, plants at the five-leaf stage during sowing date I were damaged by frost in May, while the younger plants (three-leaf stage) during the sowing II were not affected by the same frost event. Similarly, plants at the five-leaf stage during sowing III were highly affected by the July frost, while the older plants (eight-leaf stage) escaped any damage during sowing II. 

Data on frost damage to seedlings at the same stage (five-leaf stage) during sowing dates I and III were combined and analyzed to identify the G × S interaction. The number of damaged plants was significantly different for genotypes (6.7 ***, *p* <0.001), sowing dates (2.5 ***), and their interaction (9.5 **, *p* < 0.01)). The severity and frequency of mid-season frost events increased the number of damaged plants. Sowing III was exposed to frost three times for five days (3, 4, and 7 July), which led to more damaged plants than during sowing I, at a similar age (14 May). PBA Nasma showed low frost damage during both sowing dates along with 11NF010a-2, IX559d-2-2, and IX474/4-3 compared to the sensitive check Fiord (Table 1). 

#### 3.2.2. A Visual 1–9 Score on the Vegetative and Reproductive Organs

The symptoms of frost damage on stems appeared on the same day of the frost event, but one day later on flowers. Both types of damage were scored one day after the 28 July frost event. Damage was found in sowing dates I (21-leaf stage) and II (15-leaf stage), while III (nine-leaf stage) escaped the damage. 

The combined data of sowing I and II in 2015 showed a significant difference in the frost-damaged stems for genotypes (1.0 ***, *p* <0.001), sowing dates (0.4 *, *p* <0.05), and their interaction (1.4 *). Regarding flowers, the variation was found only for genotypes (0.9 ***) and the G × S interaction (1.3 *). The frost damage on stems decreased with delayed sowings. Two genotypes (IX474/4-3 and IX588d/1-4) showed low sensitivity to frost in their stems and flowers during both sowing dates, while Fiord showed maximum sensitivity to frost (Table 1).

#### 3.2.3. Frost Damage on Flowers

Flowers were tagged the day after frost occurred on 28 July 2015 during sowing dates I and II, but sowing III was excluded from this evaluation as flowering had not occurred on those plants. A significant difference was found in the survival of flowers (noted if they produced pods) for genotypes (10.3 *) and sowing dates (3.8 ***), but their interaction did not show any differences. The percentage of flower survival for sowing date I was higher than for sowing date II (Table 1). The genotypes 11NF010a-2, Fiord, and PBA Nasma were among the genotypes with the highest percentage of flower survival for both sowing dates, with over 30% for sowing I and 13% for sowing II. Flower survival in Cairo was high for sowing I (25.8%) but very low for sowing II (5.2%). Likewise, IX561f-4-2 showed a significant drop from sowing I (20%) to sowing II (1.9%). Only three genotypes (IX541a-2-8, PBA Nanu, and PBA Warda) showed similar survival percentages for both sowing dates; however, they did not exceed 12% (Table 1).

### 3.3. Frost Damage in 2016 at Narrabri

The 2016 growing season received an outrageous amount of rainfall; primarily, cloudy skies and warm nights created an unfavorable environment for frost. Therefore, only four light frosts were recorded in the season, and the damage appeared only after the fourth event, on 30 July. Symptoms of frost damage were found on stems and juvenile flowers (closed flowers) during sowing dates I and II, but none were found during sowing date III. Juvenile flowers were more sensitive to frost than fully opened flowers. A significant difference in terms of stem damage was found for genotypes and sowing dates (1.2 *** and 0.4 ***, respectively) and also in the damaged flowers (1.2 * and 0.4 *, respectively), while their interaction showed a variation only for stems (0.7 *). The frost damage on stems and flowers in 2016 was less than the previous growing season. Hence, the data were not pooled between years.

Frost damage on stems and pods in sowing I was more significant than in sowing II (Table 1). About 60% of genotypes showed more damage to stems on the first sowing date than the second, while five (IX474/4-3, 11NF014d-4, IX585c/1-11, PBA Nanu, and PBA Nasma) showed similarly low damage on both sowing dates. Three genotypes (IX541a-2-8, Doza, and IX525c-1-10) were susceptible to frost, whereas the genotypes 11NF014d-4, IX474/4-3, and IX588d/1-4 showed tolerance. Those genotypes were also less affected by frost in 2015.

### 3.4. Frost Damage and Pod Set in 2016/2017 at Dirab (Saudi Arabia)

#### 3.4.1. Response to Frost

The most common frost in Saudi Arabia is advection frost. A harsh windy frost (−4.19 °C) occurred on 5 February 2017 at Dirab. This frost damaged all flowers from all genotypes, but genotypic variation was seen on stems and pods. Frost damage was observed on all plant parts; the upper parts (such as pods) showed more damage than the lower, the opposite of what was observed after the radiation frost in Narrabri. This advection frost severely damaged plants and prevented them from continuing to produce further flowers and pods, but it did not kill the plants, and they continued to grow.

A significant difference was found among genotypes in terms of frost damage on stems (1.7 ***) and pods (1.9 *). All 16 genotypes showed higher frost damage on stems than on pods (Table 1). IX525c-1-10, PBA Warda IX474-3 and 11NF010a-2 showed the lowest frost damage on stems. PBA Warda and IX474/4-3 also showed the lowest damage on pods. Many genotypes, including Fiord, IX585c/1-11, IX541a-2-8 and IX561f/4-2, which had high sensitivity to frost on stems, showed moderate levels of tolerance on pods. IX474/4-3 and 11NF010a-2 showed similar levels of tolerance at both stages. The local variety, Giza Blanka, had higher sensitivity to frost on stems than on pods, but it was less tolerant than the Australian-bred genotypes. 

The effect of low temperature on pod set was studied by tagging flowers. The percentage of flowers forming pods differed significantly among genotypes (14.1 *) and the time of flower tagging (6.1 ***) (tag_1_ (50 DAS), tag_2_ (64 DAS) and tag_3_ (77 DAS)), but not for their interaction. However, the time taken from flowering to pod formation showed significant differences according to genotype (1.1 *), time of flower tagging (0.5 ***), and their interaction (1.9 **) (Figure 4). 

Flowers produced at the tag_1_ (50 DAS) and tag_2_ (64 DAS) stage showed a genotypic variation in terms of the percentage the flowers forming into pods. The number of flowers produced were more than double the flowers in tag_3_ (77 DAS) (Table 2). Of the 16 genotypes, six (PBA Nanu, IX474/4-3, PBA Nasma, Doza, IX585c/1-11, and 11NF014d-4) showed the highest percentage in tag_2_, two (Fiord and IX541a-2-8) had a similar percentage in tag1 and tag_2_, one (IX561f-4-2) had the lowest in tag_2_, and the remaining genotypes had the highest percentage in tag_1_.

Among the genotypes in all flower tags, IX585c/1-11 in tag_2_ showed the highest percentage of flower conversion to pods (75%). In contrast, Giza Blanka was the only genotype at tag_3_ (77 DAS) that did not produce any pods Three genotypes (IX525c-1-10, Cairo, and 11NF014d-4) produced pods at a rate above the mean from all flowers in Tag_1_, Tag_2_, and T_3_, while Giza Blanka and IX474/4-3 produced pods at a rate below the mean. Flowers differed in terms of the time taken for conversion into pods because of the effect of genotypes, time of flower formation (e.g., formed at the beginning of flowering or a month later), and their interaction. For example, flowers in IX559d-2-2 took longer to produce pods as the plant advanced; it took 14 to 15 days at tag_1_, 16 to 17 days at tag_2_, and 18 to 19 days at tag_3_ (Table 3). In addition, flowers in Fiord and IX585c/1-11 required a short time for conversion into pods in tag_1_ (13 days) but took a long time in tag_3_ (17 days). However, flowers in three genotypes (11NF010a-2, PBA Nanu, and IX541a-2-8) took the same time (15 to 16 days) in all tags. Overall, flowers in tag_1_ and tag_2_ took the shortest time to form pods, while flowers in tag_3_ took the longest (Table 3). 

#### 3.4.2. Correlation of Frost Tolerance with Phenology and Grain Yield in 2015, 2016, and 2016/2017

Plant traits differed in terms of their correlation with frost damage. Frost damage at the seedling stage (five leaf nodes) in 2015 had a negative correlation with plant height at physiological maturity (Table 4). In addition, frost during the seedling stage in 2015 positively correlated with flowering but negatively correlated with physiological maturity, seed filling duration (SFD), and yield. The damaged stems in 2015 and 2016 showed no correlation with flowering, maturity, and SFD, but had a high negative correlation with yield. The frost damage on flowers did not correlate with plant phenology, SFD, and yield. Regarding the survival of flowers in 2015, they showed a high positive correlation with plant height at 50% physiological maturity and yield (r^2^ = 0.4 ***). 

Based on these results, the frost damage on a particular part was usually correlated with the damage on another part. For example, the frost damage on stems was positively correlated with the damaged flowers in 2015/2016 and the damaged pods in 2016/2017. A high negative correlation was found between the percentage of flower forming pods and the duration of pod formation (r = −0.5 ***).

## 4. Discussion

### 4.1. Effect of Frost Damage in 2014−2016 at Narrabri, and in 2016/2017 at Dirab

The current study showed that sowing time plays an essential role in minimizing frost damage. For example, seedlings suffered the most frost damage in the case of late sowing. However, frost damage after this stage was higher at sowing dates I and II, as observed in the three consecutive growing seasons (2014, 2015, and 2016), while the damage on stems and pods was rather minor at sowing date III in 2014 and none occurred (on stems and flowers) in 2015 and 2016. According to Matthews et al. [29], avoiding very early plantings reduces the impact of frost and ultimately increases yields. However, other factors, such as drought and heat stress, can also cause significant yield losses if there is a delay in sowing. Obtaining the optimum yield of any crop depends upon the sowing time to minimize the worst risks of environmental stress. These results showed that sowing faba bean in the first week of May (sowing II) at Narrabri will reduce yield loss due to frost. 

Frost, at an early stage, damaged vegetative organs and limited the plant development. It was found that an increase in frost damage at the seedling stage (five-leaf stage) in 2015, decreased plant height at physiological maturity (r = −0.6 ***), decreased SFD (r = −0.5 ***), and ultimately decreased yield (r = −0.6 ***). Crops, including faba bean, require a long time to recover after serious frost injury, affecting growth and development, and ultimately seed yield [30,31]. Although the correlation of frost damage on pods in 2014 (Narrabri) and 2016/2017 (Dirab) was not tested with yield, the frost damage on pods was identified, which definitely would have affected the yield. The vegetative and reproductive organs are both affected together when frost occurs, but the damage differs based on their tolerance. 

Crops differ in terms of frost tolerance according to their growth stage. During the seedling stage in 2015, frost damage was observed at the five-leaf stage at sowing dates I and III, but sowing date II escaped the damage from this frost event as the plants were at the three-leaf stage and eight-leaf stage, showing that faba bean plants were vulnerable to frost in the period between the five- and seven-leaf stages. Contrary to these findings, Murray et al. [32], and Kephart and Murray (unpublished data), found a variation in frost tolerance during the seedling stage in lentil (*Lens culinaris* L.), where ten-day-old plants survived better than six-week-old plants when exposed to frost at −3 °C, −6 °C and −9 °C. These results indicated that the sensitive stage of frost damage differed between crops and growth stages. 

In the current study, frost damage occurred on all parts of the plant, but the sensitivity differed among them (from lower to higher tolerance): flowers < stems < pods (Table 1). Frost may damage the sensitive young flowers significantly if it occurs before fertilization. Although standard flowers (fully open) may suffer external frost damage, the embryo may escape damage. In the current study, Fiord showed significant external flower damage in 2015, but the flower to pod conversion was high (Table 1). Faba bean flowers were most sensitive to frost, but because of the indeterminate growth habit, occasional light frosts will not affect the final yield. This is inferred from the non-significant correlation between damaged flowers (either juvenile or standard) and grain yield over two growing seasons in Narrabri. The 2014 (Narrabri) and 2016/2017 (Dirab) experiments found that the frost damage of pods was lower than that of stems. However, younger pods were more sensitive to frost than older pods, as observed in 2014 at Narrabri. The thick wall of older pods might have made them more tolerant to frost than younger pods and other parts (e.g., stems). Similar findings were reported by Liu et al. [33], suggesting that frost was more harmful on younger pods.

The severity of frost damage on the plant’s vegetative and reproductive organs may vary based on the type of frost. In the current study, radiation frost in 2014 harmed the lower pods, while advection frost in 2016/2017 damaged the upper pods. Forming a thin layer of cold air immediately above the ground during the radiation frost caused the damage on the lower pods, but the colder wind at the upper layer, a feature of advection frost, damaged the upper pods more. The same phenomenon was observed by Rana [34] in broad beans, where the lower parts were affected more by radiation frost. In addition, the flowers that were externally damaged by radiation frost in sowing I during 2015 showed a higher conversion percentage to pods (21.4%) than in sowing II (10.9%). This may have occurred because the plants in sowing I were taller, and their flowers had a longer distance from the ground than in II, leading to less exposure to colder air at the ground level, which protected their ovules from frost damage. Therefore, a positive correlation was found between the surviving flowers and plant height at 50% physiological maturity (r = 0.4 ***).

Variation was found among the genotypes in terms of their tolerance to frost during the vegetative and reproductive stages. For example, the 2014 experiment showed that various genotypes showed low damage on stems but high damage on pods (ATC65269), significant damage on stems but low damage on pods (0674*0805#10014), and significant damage on both stems and pods (11NF014d-4 and Fiord), suggesting that frost tolerance at the vegetative stage does not relate to tolerance at the reproductive stage, as stated by Fuller et al. [6] and Maqbool et al. [14]. In addition, no correlation was observed between frost at the seedling stage (five-leaf stage) and the frost damage on stems and flowers when frost occurred at the reproductive stage in 2015. For example, IX559d-2-2 was less affected by frost at the seedling stage in 2015, but its stems and flowers (2015, 2016, and 2016/2017) suffered more at subsequent growth stages. Fuller et al. [6] had found the same phenomenon in wheat, where plants that were tolerant to −10 °C before stem elongation suffered severe damage when they were exposed to −4 °C after elongation. 

Frost tolerance seems to be strongly inherited. For example, four of the tolerant genotypes in 2014 (11NF003a-11, 11NF003a-16, 11NF003b-1, and 11NF003e-4) were sister lines with the pedigree: 354g/1-4×114/1-16. 114/1-16 is PBA Warda, which has tolerance to frost, and this was transferred to its progenies. PBA Nasma (038/1-09×004-16) had moderate to high tolerance to frost on seedlings, stems, flowers, and pods in the three growing seasons (2015, 2016, and 2016/2017). Inclusion of these varieties in the breeding program led to high frost tolerance genotypes. For example, IX474/4-3 (PBA Warda×AF04085) and 11NF010a-2 (331b/1-1×PBA Nasma), in 2015 and 2016/2017, showed low damage in relation to the seedling stages, stems, flowers and pods. PBA Nasma and 11NF010a-2 showed a high conversion percentage of flowers to pods. The genotype PBA Warda was tolerant to frost, but its flowers showed low conversion into pods, whereas, for the Fiord genotype, it was completely the other way. Therefore, it is possible to obtain new frost-tolerant genotypes with high conversion rates of flowers into pods from a cross between such varieties. Giza Blanka, the local variety in Saudi Arabia, showed high sensitivity to frost on stems and pods, whereas other lines showed tolerance. Hence, several Australian-bred varieties and elite lines have significantly better frost tolerance than the local Saudi variety. 

### 4.2. The Effect of Low Temperatures on Pod Set

The success of pod setting depends on several factors, such as genotype, the stage of flowering, and the surrounding environment. Although faba bean is an indeterminate crop, flowers produced at the beginning of flowering have high chances of developing into pods. In our studies, nearly 40% of the 768 flowers tagged at 50 DAS and 64 DAS (tag_1_ and tag_2_) converted into pods while the percentage of pods formation dramatically decreased from flowers produced at 77 DAS (tag_3_), where only 19% of the 384 flowers produced pods. None of the flowers converted into pods after 90 DAS (tag_4_) because of the frost event (Figure 4). The average rate of flowers conversion into pods for all plants during the three plant stages was 34.2%, which is much higher than 20% reported by Macfarlane et al. [22]. The difference could be because of different environmental conditions and genotypes. Although the ambient temperature at all three stages was similar—23.9/8.7 °C (day/night) in tag_1_, 25.5/7.2 °C in tag_2_, and 24.2/8.9 °C in tag_3_—the pod formation decreased substantially at 90 DAS. Several studies found similar results for legumes, specifically that most pod setting occurs soon after flowering. For example, the production of a high number of flowers at the beginning of the flowering phase (two to three weeks) led to a high pod yield in mung bean (*Vigna radiata* L.) and groundnut (*Arachis hypogaea* L.) [35,36]. In addition, a two-year field experiment (1985 and 1986) on field peas (*Pisum sativum* L.) in Western Australia showed that the first three podding nodes contributed the most to yield compared to the subsequence nodes [37]. In other studies, over 70% of pods per plant were set in the first 10–15 days after flowering in faba bean [38], soybean (*Glycine max* L.) [39,40] and pigeon pea (*Cajanus cajan* L.) [41]. To sum up, the early period of the flowering phase is the time for forming pods, and it is the most critical time in which pod sets may be most harmed by low temperatures, leading to a significant yield loss. According to Martin and Sauerborn [42], sowing faba beans in Syria (Mediterranean climate) in September or October will decrease the chance of obtaining high yields because it will expose the plants to harsh frost in December and January when they are at the beginning or middle of the flowering period. 

Time taken from flowering to pod formation is also considered a fundamental aspect of the success of pod formation. Based on the current results, pod formation was more successful when the time taken to turn flowers into pods was short. The 2016/2017 experiment showed a highly negative correlation between the percentage of flowers conversion to pods and the duration of this conversion (r = −0.5 ***). Pod setting is a necessary process in the production of grain yield, but it has an acute sensitivity to abiotic stresses such as drought and temperature extremes [43]. However, genotypes that took a shorter time to convert into pods will benefit the breeding program, as they also have a higher percentage of pod formation, which will ultimately improve yield potential. For example, flowers in IX525c-1-10 showed the highest conversion into pods in tag_1_ (54.6%) within a short period of time (13.3 days). Similarly, flowers in IX585c/1-11 had the shortest duration and highest conversion (13.5 days, 75%) in tag_2_, and IX561f-4-2 had the shortest duration and highest conversion in tag_3_ (15.3 days, 33.3%). These genotypes have the potential to increase yield. 

Three of the 16 genotypes (11NF010a-2, PBA Nanu, and IX541a-2-8) took the same amount of time to form pods (15–16 days), while other genotypes showed variations in their pod formation, indicating that a well-chosen genotype and sowing date could avoid exposure to abiotic stresses. Early pod formation and maturity are needed for short growing season environments. Two chilling-tolerant chickpea genotypes (Sonali and Rupali) released in Australia were characterized by early pod formation and maturity before the onset of water deficit [44]. Hall and Grantz [45] further showed that early pod formation could be used as an approach for the selection of drought tolerance in cowpeas (*Vigna sinensis* L.), supporting the findings of our study that reducing the risk of frost damage is possible by selecting genotypes that have faster pod formation.

## 5. Conclusions

A high level of variation for frost tolerance was observed among the genotypes in this four-year experiment. However, the 2016 results need to be taken with caution because of the low frost severity, requiring further testing for conformity, especially for tolerance at the reproductive stage. Delayed sowing was shown to enable the avoidance/minimization of frost damage, but it will also decrease overall yield. The growth stages showed differences in terms of their sensitivity to frost, e.g., the five-leaf stage was more sensitive to frost than the three and seven-leaf stages. The lower plant part, where most pods are produced, was more sensitive to radiation frost, whereas the upper part was more sensitive to advection frost. Radiation frost forms immediately above the ground; therefore, flowers of taller plants have a better opportunity to avoid the damage of radiation frost because of their long distance to the ground. Flowers produced during the initial flowering period set more pods than those produced later. Therefore, a severe frost occurrence during the early stage of flowering can cause a significant yield loss. 11NF010a-2 showed a high level of frost tolerance at the seedling, flowering, and pod development stages; meanwhile, it also showed a high rate of flower conversion to pods. Other frost tolerant genotypes found were IX474/4-3, PBA Warda, and PBA Nasma, whereas IX525c-1-10, IX585c/1-11, and IX561f-4-2 took the shortest time to covert from flowers into pods, and also had the highest flower conversion rate to pods. Inclusion of these genotypes in breeding programs can increase yields as well as frost tolerance. 

## Figures and Tables

**Figure 1 plants-10-01925-f001:**
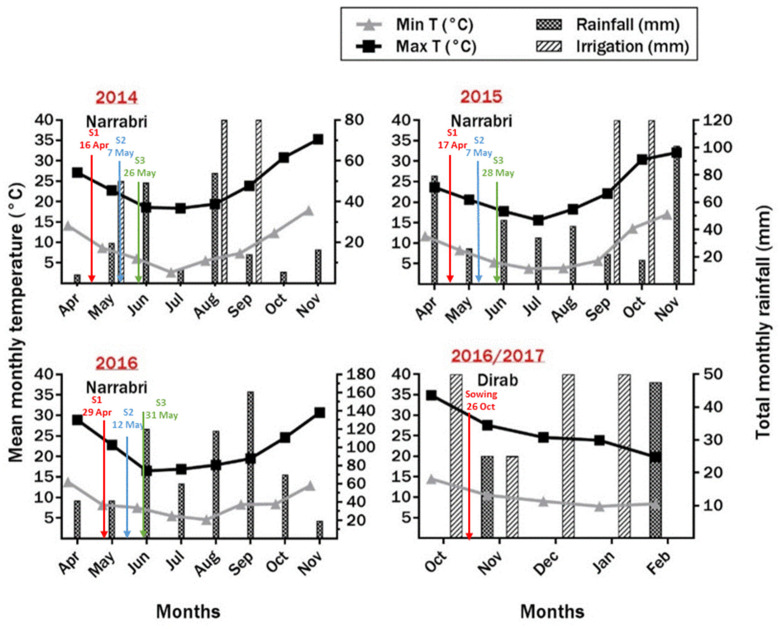
Date of sowing dates (sowing 1 (S1), sowing 2 (S2) and sowing 3 (S3)), mean of maximum and minimum temperatures, and total rainfall and irrigation, during the three growing seasons at the University of Sydney (Narrabri) and one season at the Agricultural Research Station (Dirab).

**Figure 2 plants-10-01925-f002:**
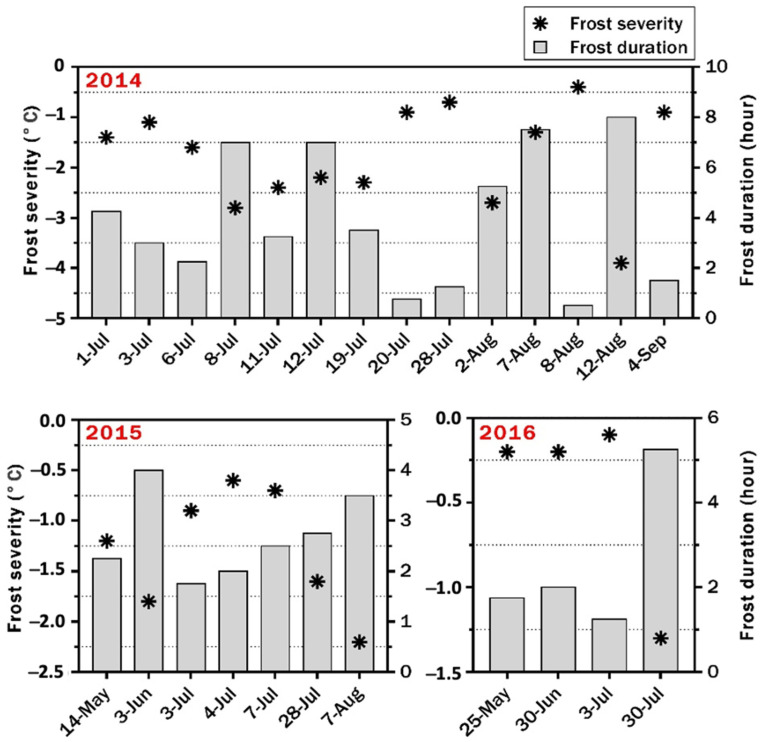
The severity and duration of frost events during the 2014, 2015, and 2016 growing seasons at the University of Sydney at Narrabri.

**Figure 3 plants-10-01925-f003:**
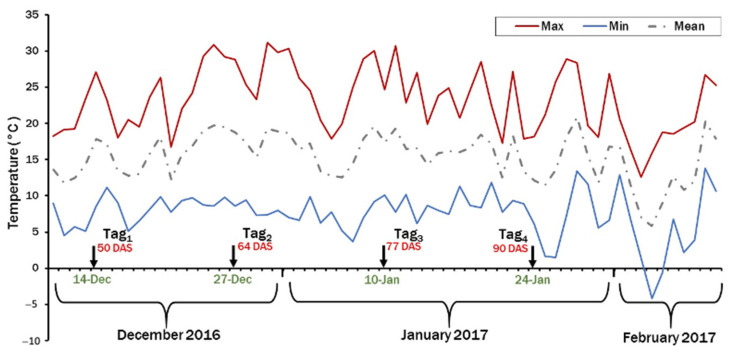
Maximum, minimum and mean temperatures at the four tags during the 2016/2017 growing season at the Agricultural Research Station at Dirab.

**Figure 4 plants-10-01925-f004:**
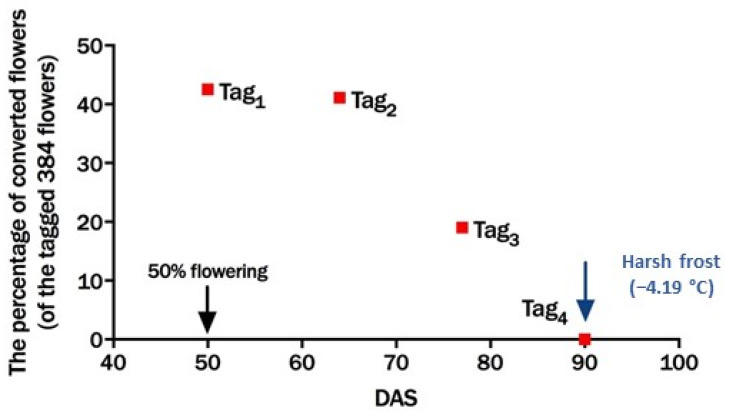
The percentage of pod formation from the tags at 50, 64, 77, and 90 days after sowing (DAS) in 2016/2017 at the Agricultural Research Station at Dirab.

**Table 1 plants-10-01925-t001:** Frost damage at different plant growth stages in 2015 and 2016 at the University of Sydney (Narrabri, Australia), and in 2016/2017 at the Agricultural Research Station (Dirab, Saudi Arabia), *n* = 3. S I, S II, and S III indicate the three sowing dates. Frost-E indicates the number of frost affected plants. Frost-S, Frost-F, and Frost-P indicate frost damage on stems, flowers, and pods, respectively. Survival-F is the percentage of frost affected flowers that produced pods. Frost-E was evaluated by counting the number of damaged plants 20 m^−2^, while Frost-S, Frost-F, and Frost-P were determined using a visual 1–9 scale.

	Location	Narrabri												Dirab	
	Year	2015								2016				2016/2017	
	Frost Damage	Frost-E		Frost-S		Frost-F		Survival-F		Frost-S		Frost-F		Frost-S	Frost-P
	Sowing Date	S I	S III	S I	S II	S I	S II	S I	S II	S I	S II	S I	S II		
**Genotypes**	IX525c-1-10	9	11	6	8	6	7	23.1	13.7	5	2	3	2	4	5
	Fiord	5	10	5	6	6	5	31	15.8	4	3	4	2	8	4
	IX588d/1-4	5	19	3	3	3	5	19.8	5.8	1	2	3	1	7	6
	Doza	4	9	4	5	6	5	17.2	12.9	5	2	2	2	6	4
	IX585c/1-11	4	35	5	5	6	7	26	9.4	2	2	1	2	8	4
	Cairo	3	12	4	6	4	4	25.8	5.2	4	2	3	2	7	5
	11NF010a-2	2	5	3	2	4	3	33.3	13.3	3	2	1	1	4	4
	IX541a-2-8	2	10	7	7	6	7	10.4	10	6	3	3	4	7	4
	PBA Warda	2	10	4	5	4	4	8	11.5	4	2	3	2	3	2
	11NF014d-4	1	24	4	2	4	3	24.8	17.7	2	1	1	2	7	5
	IX474/4-3	1	8	3	2	2	2	24.4	10.8	1	2	2	1	4	3
	PBA Nanu	1	12	4	4	6	6	9.7	9.9	3	3	3	3	6	4
	IX559d-2-2	1	7	4	6	5	5	16.4	10.2	4	4	3	2	7	5
	IX561f-4-2	0	12	4	5	4	5	20	1.9	3	2	2	1	7	4
	PBA Nasma	0	5	5	5	5	5	30.8	15.6	3	3	1	2	5	5
	Giza Blanka	-	-	-	-	-	-	-	-	-	-	-	-	7	5
	Mean	2.7	12.6	4.3	4.7	4.7	4.9	21.4	10.9	3.3	2.3	2.3	1.9	6.1	4.3

**Table 2 plants-10-01925-t002:** Percentage of flowers converted to pods at three stages of faba ban (Tag_1_, Tag_2_ and Tag_3_) in 2016/2017 at the Agricultural Research Station at Dirab (*n* = 3). Means denoted by different capital letters are significantly different according to time of tagging (l.s.d. = 6.1 ***), and small letters denote genotypes within each time of tagging (l.s.d. = 14.1 *).

No.	Genotypes	Tag_1_			Tag_2_			Tag_3_		
1	IX588d/1-4	58.3	A	a	37.5	B	bcd	12.5	C	cde
2	IX559d-2-2	56.5	A	ab	29.2	B	cde	25	B	abc
3	IX525c-1-10	54.6	A	ab	45.8	B	b	29.2	C	ab
4	11NF010a-2	50	A	abc	37.5	B	bcd	16.7	C	bcd
5	Cairo	50	A	abc	41.7	B	bc	25	C	abc
6	Fiord	50	A	abc	50	A	b	16.7	B	bcd
7	PBA Warda	50	A	abc	41.7	B	bc	8.3	C	de
8	IX585c/1-11	47.2	B	abcd	75	A	a	18.8	C	bcd
9	11NF014d-4	43.5	B	bcde	50	A	b	20.8	C	abcd
10	IX541a-2-8	36.1	A	cdef	41.7	A	bc	20.8	B	abcd
11	Doza	35.2	B	def	45.8	A	b	12.5	C	cde
12	IX561f-4-2	33.2	A	def	25	B	de	33.3	A	a
13	PBA Nasma	30.4	B	ef	41.7	A	bc	20.8	C	abcd
14	Giza blanka	29.2	A	f	20.8	B	e	0	C	e
15	PBA Nanu	28.9	B	f	37.5	A	bcd	25	B	abc
16	IX474/4-3	26.7	B	f	37.5	A	bcd	18.8	C	bcd
	Mean	42.5	A		41.1	A		19.0	B	

DAS: days after sowing. * and *** significant difference at 0.05 and 0.001, respectively. *n* is the number of replications.

**Table 3 plants-10-01925-t003:** Time taken from flowers to produce pods in all tags (Tag_1_, Tag_2_ and Tag_3_) in 2016/2017 at the Agricultural Research Station at Dirab (*n* = 3). Means denoted by different capital letters are significantly different among time of tagging (l.s.d. = 0.5 ***), and small letters among genotypes within each time of tagging (l.s.d. = 1.1 *). Bold/underline genotypes show those where they took 15−16 days to produce pods from the tagged flowers.

No.	Genotypes	Tag_1_			Tag_2_			Tag_3_		
1	PBA Nasma	16.7	B	a	15.0	C	bcd	17.3	A	abc
2	IX561f-4-2	16.3	A	ab	15.3	B	bcd	15.3	B	ef
3	** PBA Nanu **	16.0	A	abc	15.3	B	bcd	15.7	AB	def
4	11NF014d-4	15.3	B	bcd	15.3	B	bcd	16.7	A	bcd
5	PBA Warda	15.3	B	bcd	15.0	B	bcd	17.0	A	abc
6	** 11NF010a-2 **	15.0	B	cde	15.7	A	abc	15.3	AB	ef
7	** IX541a-2-8 **	15.0	A	cde	15.0	A	bcd	15.0	A	f
8	IX559d-2-2	14.7	C	def	16.7	B	a	18.0	A	a
9	IX588d/1-4	14.7	B	def	14.3	B	de	17.0	A	abc
10	Cairo	14.3	B	defg	14.7	B	cd	16.3	A	cde
11	Doza	14.0	B	efgh	14.3	B	de	17.0	A	abc
12	Giza blanka	13.7	C	fgh	16.0	B	ab	16.9	A	abc
13	IX474/4-3	13.5	C	gh	14.5	B	de	17.5	A	ab
14	IX525c-1-10	13.3	C	gh	14.7	B	cd	15.7	A	def
15	Fiord	13.0	C	h	14.7	B	cd	17.3	A	abc
16	IX585c/1-11	13.0	B	h	13.5	B	e	17.0	A	abc
	Mean	14.6	B		15.0	B		16.6	A	

DAS: days after sowing. * and *** significant difference at 0.05 and 0.001, respectively. *n* is the number of replications.

**Table 4 plants-10-01925-t004:** Pearson correlation coefficients of frost damage in relation to seedling stage (Frost-E), stems (Frost-S), and flowers (Frost-F), with phenology, plant height at 50% physiological maturity, and yield, in 2015 and 2016, at the University of Sydney at Narrabri (*n* = 15).

2015—Narrabri		Frost-E	Frost-S	Frost-F	Flowering	Maturity	SFD	Height	Yield
	Frost-E	-							
	Frost-S	n.s.	-						
	Frost-F	n.s.	0.5 ***	-					
	Flowering	0.4 ***	n.s.	n.s.	-				
	Maturity	−0.5 ***	n.s.	n.s.	−0.7 ***	-			
	SFD	−0.5 ***	n.s.	n.s.	−0.9 ***	0.9 ***	-		
	Height	−0.6 ***	n.s.	n.s.	−0.6 ***	0.8 ***	0.8 ***	-	
	Yield	−0.6 ***	−0.3**	n.s.	−0.6 ***	0.7 ***	0.7 ***	0.6 ***	-
2016—Narrabri		Frost-S	Frost-F	Flowering	Maturity	SFD	Yield		
	Frost-S	-							
	Frost-F	0.3 **	-						
	Flowering	n.s.	n.s.	-					
	Maturity	n.s.	n.s.	0.5 ***	-				
	SFD	n.s.	n.s.	−0.9 ***	n.s.	-			
	Yield	−0.3 **	n.s.	0.3 **	n.s.	−0.3 **	-		

n.s., not significant; ** and *** significant difference at 0.01 and 0.001, respectively. *n* is the number of genotypes.

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
