# Peer review of "Evaluation of Frost Damage and Pod Set in Faba Bean (Vicia faba L.) under Field Conditions"

_plants, 2021, doi:10.3390/plants10091925_

Round 1

Reviewer 1 Report

The study is sound. As a matter of fact, I like the idea of a combined study on radiation frost and advection frost. In my opinion, this aspect could be made more clear in the conclusion. Are there cultivar differences (of those tested), which would perform better under the one (or oher) kind of stress? That is to say: Require different kinds of frost different selection/breeding strategies?

The language is appropriate. However, I had the impression that the style of the discussion chapter was worse.

The most negative aspect is that the citation stye and references do not follow the author guidelines at all. Either the authors have not appreciated the journal adequately, or the manuscript has already been submitted elsewhere.

Smaller aspects, I have mentioned within the text.

Author Response

Dear reviewer,

Thank you for your recommendations, which will contribute to the improvement of the manuscript. All changes are explained in the attached file.

Reviewer 2 Report

General: This is an interesting study examining faba bean cultivars tolerance to frost damage. However, there are some procedural items that could immensely improve the clarity of the data. It seems it would help if the authors normalized this data by pegging dates of sowing, frost dates, and data collection to growing degree days. It is tedious to read about different data collection at the various sites so many days after sowing when the authors and readers could detect trends better if damage occurred at one site a x GDD and the other site at y GDD. It might allow combining the data and definitely to model what is happening. As is, it is just observations without a lot of synthesis. 

Another issue. I could not find yield data. There is a yield column in Table 4 but it is blank. Without yield data, this is a purely observational study which does not provide any objective data.

Specific:

The tables are confusing to read.

Table 1 - what are the units of the column headers? I do not see any way to compare stats here either. 

Table 2 and 3 - Column headers need units. Difficult within the table to see what TAG1-3 are. Not easy to see what the small cap and large cap mean separation letters refer to. Tables should stand on their own without going to text numerous times.

Table 4 - Are these significance figures from F-test? This is the only place in the manuscript I see a reference to yield and it is blank. Leave out the yield column if there are no yield data reported which is problematic as well.

Author Response

(The authors gave the same response as above.)

Reviewer 3 Report

Dear Authors,

Your manuscript corresponds to the subject of the journal and can be accepted after minor revision. Below you will find my comments on the text.

L 20 Add a little more of the results obtained to the abstract before discussing and summarizing them.

L 67-68, 377, 459, 461, 464, 465, 493 and in all text - Use italics to indicate the Latin names of plant species

L 81-83 Please, add the hypothesis of your research. This will allow the reader to better evaluate your manuscript.

L 126 - Please, describe frost damage scale more clearly. May be you have some photos?

L 152 - Fig. 3

L 164-347 I would recommend changing the approach to presenting the results. Now they sometimes look like a discussion. Show more actual data, as well as the results of their statistical processing in the form of tables and graphs. So far, the results are presented somewhat chaotically, which reduces the scientific level of the manuscript.

L 348-508 I guess that after the formulation of the research hypothesis, the sections Discussion and Conclusion could be partially edited.

Author Response

(The authors gave the same response as above.)

Round 2

Reviewer 2 Report

Edits are acceptable.